# Improved Phenolic Profile, Sensory Acceptability, and Storage Stability of Strawberry Decoction Beverages Added with Blueberry Decoctions

**DOI:** 10.3390/molecules28062496

**Published:** 2023-03-09

**Authors:** Ana María Sotelo-González, Iza Fernanda Pérez-Ramírez, Julissa Haydee Soto-Infante, Haiku Daniel de Jesús Gómez-Velázquez, Ma. Estela Vázquez-Barrios, Alexandro Escobar-Ortíz, Rosalía Reynoso-Camacho

**Affiliations:** 1Chemistry School, Universidad Autónoma de Querétaro, Querétaro 76010, Mexico; 2Institute of Neurobiology, Universidad Nacional Autónoma de Mexico Campus Juriquilla, Querétaro 76230, Mexico

**Keywords:** strawberry, blueberry, beverages, stability, polyphenols

## Abstract

Blueberries are rich in polyphenols, anthocyanins, and proanthocyanidins; however, they are expensive. In contrast, strawberries have a lower cost and are rich in ellagitannins. Therefore, a strawberry–blueberry blend decoction could produce a low-cost beverage with a rich and diverse phytochemical profile. In this study, we developed three berry-based beverages: blend strawberry–blueberry (SBB), strawberry (SB), and blueberry (BB). The polyphenol profile of the beverages was characterized by UPLC-ESI-Q-ToF MS^E^, an acceptability test was carried out with potential consumers, and a stability analysis was performed under commercial storage conditions (4, 25, and 34 °C). The SBB exhibited a good sensorial preference (score of 81.3) and showed high contents and a diverse composition of anthocyanins and proanthocyanidins, which were up to 3.0- and 1.2-fold higher compared to the SB, respectively. Regarding the storage stability, the SBB showed better retention of lightness (97.9%) and red color (66.7%) at the end of the storage under refrigerated conditions (4 °C) compared to the SB. Therefore, these results demonstrate that using blended berry fruits leads to the creation of a functional beverage that has higher nutraceutical potential than single-berry-based beverages.

## 1. Introduction

The global consumption of sugar-sweetened beverages is associated with adverse health conditions, including an increased risk of cardiometabolic diseases. An alternative is the consumption of low-calorie functional beverages, such as herbal teas that are usually prepared as decoctions, a process that allows the extraction of the bioactive compounds that are responsible for their beneficial effects. Berries are the most widely used fruit for the elaboration of fruit teas due to their low caloric content, their high sensory acceptance, and their high content of bioactive compounds, such as polyphenolic compounds [1].

Strawberries (*Fragaria* spp.) and blueberries (*Vaccinium* spp.) are considered berries with high commercial value, high global demand, and numerous health benefits associated with their high content of polyphenols [2]. Specifically, blueberries have a higher concentration of polyphenols than strawberries (1.77-fold), flavonoids (6.89-fold), proanthocyanidins (3.25-fold), and anthocyanins (5.54-fold) [3]. In contrast, strawberries are richer in ellagitannins (71–83 mg/100 g), such as casuarictin, pedunculagin, and sanguiin H6 [4].

Regrettably, blueberry prices have significantly augmented worldwide in the last few years, reaching a 45% annual increase [5], thus representing a disadvantage for the production of blueberry-based functional beverages. In addition, it is noteworthy that berry-based beverages show a marked loss of their red, blue, or purple colors during storage due to their anthocyanin contents, which are highly sensitive to external factors such as alkaline pH and high temperatures [6]. Nevertheless, it has been reported that the combination of different anthocyanin profiles can improve color retention and anthocyanin stability, thus maintaining their antioxidant properties [7].

Several strategies are used to develop beverages rich in bioactive compounds with sensory attributes that are attractive to potential consumers. Among these approaches is the exhaustive chemical characterization of the constituents associated with health-beneficial effects. In this regard, berries are rich in polyphenols that are commonly identified and quantified by liquid chromatography–mass spectrometry (LC-MS) techniques. However, since polyphenols show low stability under storage conditions, including light and temperature, evaluating their content during storage is highly recommended, including the physicochemical characteristics associated with these compounds, such as color and acidity. On the other hand, sensory attributes are commonly assessed by consumer acceptability tests with a hedonic scale, which allows for predicting the consumer acceptance of the product and providing data that allow reformulation if necessary [8]. Therefore, the aim of this study was to elaborate and characterize a sensorially accepted blended berry-based beverage with greater polyphenol diversity and stability during storage compared to strawberry- or blueberry-based beverages.

## 2. Results

### 2.1. Polyphenol Profile by the UPLC-ESI-Q-ToF MS of Berry-Based Beverages

The UPLC-ESI-Q-ToF MS^E^ analysis allowed the identification and quantification of sixteen polyphenols in the SB, nineteen in the BB, and twenty-two in the SBB, for which their identity was confirmed by a comparison of their exact mass with a mass error of <5 ppm, their fragmentation pattern, and their isotope distribution. Anthocyanins were the major compounds identified in all berry-based beverages, where the strawberry beverage (SB) showed 17 μg/mL of total anthocyanins, which included pelargonidin hexoside as the major compound. The blueberry beverage (BB) contained 100.6 μg/mL, which showed a rich anthocyanin profile with the following major components: malvidin hexoside > petunidin hexoside > peonidin hexoside > delphinidin hexoside. Finally, the total anthocyanin content of the strawberry–blueberry blend beverage (SBB) was 57.5 μg/mL, which was higher than that found in the SB due to the high anthocyanin content of the BB (Table 1). Additionally, delphinidin hexoside was not detected in the SB but was quantified in the SBB. Accordingly, a mixed profile of anthocyanins was identified in this blended beverage.

A wide variety of non-pigmented flavonoids were identified in all berry-based beverages, including flavanols and flavonols; the latter were found in greater concentrations in the blueberry- and blended berry-based beverages as compared to the strawberry-based beverage (7.1–12.8-fold). Quercetin hexoside was the major non-pigmented flavonoid identified in BB and SBB (0.696 and 0.387 mg/100 mL, respectively). On the other hand, SB showed the highest concentration of hydroxycinnamic acids (3.1 μg/mL), such as coumaric acid hexoside and ellagic acid. Interestingly, the SBB generated a beverage with a 6 times higher concentration of total hydroxycinnamic acids (1.08 μg/mL) than the BB (0.18 μg/mL). On the other hand, the SBB retained 36% of pedunculagin and 61% of castalin from the BB. 

### 2.2. Consumer Acceptability Test of Berry-Based Beverages

The results of the sensory evaluation of berry-based beverages are shown in Table 2. The color of all beverages showed a high acceptability by the consumers since the histograms indicated a positive hedonic asymmetry. More than 78% of the consumers scored an SB from 7 to 9, whereas the SBB showed a mean value of 7.8 from 98% of consumers; however, no significant differences were observed in the mean score values (Table 2). The aroma was evaluated with the lowest score by the consumers (6.48–6.62); moreover, this parameter showed a moderate hedonic asymmetry, where 54% of the consumers selected a score from 5 to 7 for the SB, whereas about 64% of the consumers gave that score for the BB and SBB. 

The flavor of the three berry-based beverages recorded an average value of 7.03 (Table 2), where the BB showed the lowest rate (6.74), and the SBB had the best rate (7.28); however, no significant differences were observed. The hedonic asymmetry showed a positive trend in all berry-based beverages; however, the SB registered more dispersion since it obtained ratings from 3 to 9, where 78% of consumers scored this beverage from 6 to 9. A similar trend was observed with the BB, whereas 76% of the consumers rated the flavor of the SBB from 7 to 9 points. 

Finally, concerning the overall acceptability of berry-based beverages, the results showed a high positive asymmetry, and at least 68% of consumers scored all beverages with a score of >7. The acceptance index was calculated from these results (Appendix A). The SBB showed the highest value (81.3%), followed by the SB at 79.5% and finally by the BB at 78.0% (Table 2), indicating that all berry-based beverages were sensorially acceptable according to the rejection acceptance index of <70%, but no significant differences were observed between the mean score values of each beverage. It is noteworthy that all berry-based beverages showed negative skewness values in all sensory parameters (positive asymmetry) and low kurtosis values, where the data distributions showed a more significant flattening in these scales (Appendix A and Table 2). 

### 2.3. Stability of Berry-Based Beverages during Storage under Commercial Conditions

#### 2.3.1. pH, Titratable Acidity, and Total Soluble Solids

The physicochemical parameters of the berry-based beverages stored at 4 °C for 90 days and at 25 °C and 34 °C for 12 days are shown in Appendix A. All beverages showed low pH values, which ranged from 3.51 to 3.57. The titratable acidity of all berry-based beverages ranged between 0.18 and 0.36 g eq. citric acid/100 mL. It is noteworthy that no changes were observed in pH and titratable acidity during the 90 days of storage at 4 °C. Nevertheless, the pH values significantly decreased after 12 days of storage at 25 and 34 °C, whereas the titratable acidity was not affected at these storage temperatures. 

#### 2.3.2. Color

The SB presented the highest lightness value (L*) among all berry-based beverages (76.6), followed by the SBB (44.5) and lastly the BB (36.4). Lightness was the color parameter that showed a lower degradation rate at 4 °C for 90 days. Interestingly, despite the highest lightness value that was observed in the SB, the SBB and BB showed the highest retention of this parameter during storage at 4 °C for 90 days (97.9 and 91.2%, respectively; Figure 1A).

As expected, the BB showed a higher a* value as compared to the SB (47.3 vs. 18.3), whilst the SBB showed a mean value within these two beverages. Interestingly, even though the BB showed low lightness stability during storage, it showed the highest retention of redness under all storage temperatures (63.8–82.8%), whereas the SB showed lower retention values (33.9–61.6%). The combination of the strawberry and blueberry decoction (SBB) showed redness retention from 54.5% to 72.3%, suggesting a synergistic protective effect on SB anthocyanins (Figure 1B).

All berry-based beverages showed values of ΔE > 3.0 after only three days of storage at 25 and 34 °C and after fifteen days of storage at 4 °C (Figure 1C). Interestingly, the SBB showed a ΔE value that was 1.7-fold lower as compared to the BB after 90 days of storage at 4 °C, suggesting a lower overall color loss; nevertheless, this trend was not observed during storage at 25 and 34 °C. Finally, regarding the browning index (polymeric anthocyanins), this parameter increased in all berry-based beverages during storage, regardless of the storage temperature.

#### 2.3.3. Polyphenolic Compounds Content

The degradation kinetics of the total polyphenols of berry-based beverages stored at 4, 25, and 34 °C for 90 and 12 days are shown in Figure 2. The SB showed the most significant retention of total polyphenols under all storage temperatures (82.6–91%). The retention of the SBB was 29.8% greater than the BB after 90 days of storage at 4 °C and 9.6% greater after 12 days of storage at 34 °C (Figure 2A). Regarding total monomeric anthocyanins (Figure 2B), the SB showed only a 32.4% retention of these pigments at 4 °C, whereas the BB and SBB showed 83.3% and 85.7%, respectively. These results demonstrate that blueberries are more stable during storage than those from strawberry anthocyanins. Furthermore, the BB and SBB showed a similar anthocyanin retention at 25 °C (38.6% and 36.2%, respectively), whereas this retention decreased to 20.2% and 13.4%, respectively, when beverages were stored at 34 °C.

Regarding the total proanthocyanidins, these compounds were highly susceptible to degradation during the storage of the SB at 4 °C, obtaining a retention value of only 16.1%. In contrast, the BB proanthocyanidins were more conserved (56.3%) under the same conditions. Interestingly, the SBB conserved up to 70.3% of total proanthocyanidins at 4 °C (Figure 2C). The degradation of total polyphenolic compounds in all berry-based beverages followed a first-order reaction kinetic model, which was determined by its correlation coefficient (*R*^2^ > 0.82; Table 3). The *k* values increased in all berry-based beverages as the storage temperature augmented. The SB stored at 4 °C showed the most prolonged half-life (t_1/2_) of total polyphenols (396 days, about 13 months), which was significantly lower at 25 °C (143 days, about 5 months). Both beverages with blueberries showed lower half-life values as compared to the beverage with strawberries; nevertheless, the blended beverage showed higher values than the 100% blueberry beverage (190 vs. 74 days). Low half-life values of all polyphenol classes were observed in all berry-based beverages at high storage temperatures.

The BB showed a higher total anthocyanin content and a lower degradation rate during storage, and its half-life was equivalent to 7 months at 4 °C, whereas it was equivalent to 4.2 months for the SBB and only 2 months for the SB, which further decreased with higher storage temperatures. Finally, *R*^2^ values > 0.84 were observed for all reaction kinetics of the total proanthocyanidins of the three berry-based beverages. Interestingly, all berry-based beverages showed the lowest *k* values at 4 °C, and SBB showed the highest t_1/2_ value (114 days) as compared to single fruit-based beverages. 

### 2.4. Correlations between the Polyphenol Profile of Berry-Based Beverages and Their Degradation during Storage under Commercial Conditions

A strong positive correlation was observed between the pelargonidin hexoside, the major anthocyanin of the SB, the degradation of total monomeric anthocyanins (TMA), the loss of redness, and the increased browning index during storage (Figure 3). On the other hand, the major anthocyanins of the BB, such as malvidin hexoside, peonidin hexoside, and petunidin hexoside, were positively correlated with an increased browning index and the degradation of total polyphenolic compounds, whereas delphinidin hexoside was positively correlated with a loss of luminosity and overall color change. Finally, pedunculagin ellagitannin, detected in higher amounts in the SB, was positively correlated with the loss of redness and the degradation of monomeric anthocyanins and total proanthocyanidins.

## 3. Discussion

Plant polyphenols have received great interest in the last decade from both consumers and food-processing industries due to their health benefits; thus, they have been considered as functional ingredients in food and beverage products. However, their stability during food processing and storage is a major concern. Therefore, the identification of polyphenols associated with the stability of pigmented bioactive compounds, such as anthocyanins, could be a strategy for identifying the potential of ingredients that can be used for the production of polyphenol-rich and stable functional foods and beverages [1]. 

Pelargonidin 3-*O*-glucoside, pelargonidin 3-*O*-rutinoside, and cyanidin 3-*O*-glucoside represent more than 95% of the total anthocyanins identified in different varieties of strawberry fruit, and the anthocyanin content of strawberry juices ranges from 40 to 114 mg/L [3]. Therefore, the total content of monomeric anthocyanins in our beverages is lower than that reported for berry juices, which could be related to a dilution effect and their thermal degradation during fruit-drying and decoction processes. In addition, chlorogenic acid has been identified as the major phenolic acid of blueberry fruits; however, this compound was not identified in this study. In a previous study, we identified chlorogenic acid as the major phenolic acid of blueberry decoctions (6.97 μg/mL) using the same UPLC-ESI-QToF MSE method as used for this study [9]. Therefore, we hypothesized that the absence of chlorogenic acid in the blueberry-based beverage could be due to its thermal degradation during the pasteurization process.

Ellagitannins are the most abundant polyphenols in strawberry fruits, along with anthocyanins and flavanols, whereas blueberry fruits are widely known for their high content and variety of anthocyanins. Interestingly, in this study, the combination of both berry fruit decoctions allowed the production of a beverage with a relatively high content of all these classes of polyphenols. Similarly, Bobinaitė et al. [4] reported that the enrichment of raspberry extracts with fruit puree (pears, apples, black currants, and yellow cherry plums) increased the concentration of ellagitannin, anthocyanins, and ellagic acid in the blended product.

The differential polyphenol profile of the beverages developed in this study could be associated with the specific sensory characteristics that influenced their sensory acceptability by consumers. It is well-known that anthocyanins are responsible for the typical red, purple, or blue color of berry fruits. The aglycone, which is the central part of the structure of anthocyanins, corresponds to a flavylium cation that contains a different number of hydroxy and methoxy groups in the B-ring, which defines the final coloration of each anthocyanin, whereas the combination of all different anthocyanins and their proportion defines the final color of the berry fruit [10].

On the other hand, the consumers that participated in the sensory analysis described a subtle aroma in all berry-based beverages, which could be related to the thermal degradation of volatile organic compounds during the drying and decoction process [11]. Most consumers preferred the SBB flavor, which could be explained by the sweet and bitter flavor profile of the strawberry decoction in combination with the astringent flavor of the blueberry decoction [2].

On the other hand, all beverages showed pH values ranging within a very acidic level (pH < 4.5), which favors their storage due to the inhibition of bacterial growth [12]. Nevertheless, an increased amount of mesophilic aerobic bacteria was observed after 15 days of storage at 25 and 34 °C, indicating that the citric acid added as a pH regulator and preservative was not enough to augment the microbiological shelf life of the berry-based beverages developed in this study [13]. The increased count of mesophilic aerobic bacteria may have contributed to the decreased pH and ºBrix values at the end of storage.

Regarding the color stability of berry-based beverages during storage at commercial conditions, the combination of the anthocyanins of both berry fruits led to increased color stability in the SBB. These results agree with those reported by Grobelna et al. [7], who reported that anthocyanins from different fruits improved the red color stability in juices. For example, a mixture of 10 and 30% of honeysuckle berry juice with apple juice maintained 95.13% of the initial value of a* and 92.64% after four months of storage at 20 °C as compared with 100% apple juice (71.42%) or 100% of honeysuckle berry juice (82.68%). Nevertheless, in our study, the combination of both berry fruits did not prevent an increased browning index during the storage of the SBB. The browning index estimates the degradation of anthocyanins that lead to brown polymerized anthocyanin–tannin complexes [14]. Therefore, the high browning index of the SB indicates that this beverage develops a brown color during its storage independent of the storage temperature, which is a major disadvantage for the production of strawberry-based products since consumers clearly associate this berry fruit with a vibrant red color.

In comparison with commercial teas [15], the berry-based beverages developed in this study show a greater polyphenol concentration, up to 77.5% of total polyphenolic compounds and 19.4% of total anthocyanins, suggesting a higher nutraceutical potential. A study carried out with clear and cloudy strawberry juices from different varieties reported a mean proanthocyanidin content of 0.32–0.45 mg/mL [16], which was 2.5 times higher than the SB (0.14 mg/mL). On the other hand, Rodríguez-Daza et al., [17] reported that blueberry juices showed 32.9–36.4 mg/mL of total proanthocyanidins. These differences could be attributed to the different berry varieties, agronomic conditions, and the extraction process (juicing vs. decoction) since the interaction with husk and membranes could enrich these proanthocyanidins in the juice as compared with the decoction process.

The lower degradation rate of total polyphenols from the SB and SBB was partly related to their ellagitannin content, mainly pedunculagin, which is an important bioactive compound found in strawberry fruits. Moreover, it has been reported that raspberry ellagitannins have low degradation rates under acidic conditions [18], such as those found in our berry-based beverages. On the other hand, blueberry anthocyanins were highly susceptible to thermal degradation during storage. Few comparative studies have been reported regarding the stability of berry anthocyanins; nevertheless, Oszmiański and Wojdyło [16] suggested that the pectin contained in strawberry juices may protect the degradation rate of polyphenols during their storage at both low and high temperatures. Therefore, we hypothesized that the high degradation rates of our beverages could be partially related to the low content of pectin and other components of dietary fiber, since these high-molecular-weight compounds were poorly extracted during the decoction process.

The understanding of the kinetic parameters related to changes in the nutritional, nutraceutical, and physicochemical parameters during the storage of functional foods and beverages is essential for the prediction of their potential bioactivity. In this regard, it has been reported that some strawberry polyphenols, such as *p*-coumaric acid, improve the half-life of other polyphenols from strawberry juice [19]. This acid was found in the SB (1.32 μg/mL) and was absent in the BB, which could have contributed to the lower stability of total polyphenols in the latter. However, in this study, we identified several individual anthocyanins that were highly correlated with the overall color degradation during the storage of berry-based beverages. Interestingly, the combined anthocyanin profile of the SBB could be associated with higher anthocyanin, proanthocyanidin, and color stability during storage as compared to single-berry-fruit beverages.

## 4. Materials and Methods

### 4.1. Decoction Berries Preparation

Strawberries and blueberries were acquired from a local market in Querétaro, Querétaro, México. First, the berries were washed and disinfected with a 100 ppm NaClO aqueous solution, and they were sliced into 0.5 cm sections and dried in a forced-air circulation oven (BF 400, Binder Inc., Bohemia, NY, USA) at 45 °C for 48 h to 72 h. Then, decoctions were prepared using 100 g of each dried fruit with 1 L of boiling water (95 °C) for 15 min. Finally, berry fruit decoctions were filtered and adjusted to a volume of 1 L each.

### 4.2. Berry Beverages Preparation

Beverages were made from strawberry–blueberry 50%-50% (SBB) decoctions, strawberry 100% (SB), and blueberry 100% (BB). Sucralose (2%) and acesulfame K (2%) were added as non-caloric sweeteners. Citric acid (1%) was added for flavor modification, and potassium sorbate (E-202) (1%) was added as a preservative. Beverages were pasteurized under standard conditions (73 °C for 15 s).

### 4.3. Polyphenol Profile by UPLC-ESI-Q-ToF MS

Beverages were filtered through a 0.45 μm PVDF syringe filter. Then, samples (2 μL) were injected into a BEH Acquity C18 column (2.1 × 100 mm, 1.7 μm) in an ultra-performance liquid chromatography system coupled to a quadrupole/time-of-flight mass spectrometer with an electrospray ionization interphase (UPLC-ESI-QToF MS^E^, Vion, Waters Co, Milford, MA, USA) following the procedure reported by Reynoso-Camacho et al. [9]. Beverages were filtered through a 0.45 μm PVDF syringe filter and injected (2 μL) into a BEH Acquity C18 column (2.1 × 100 mm, 1.7 μm at 35 °C). The mobile phase was (A) water-acidified with 0.1% (*v*/*v*) formic acid and (B) acetonitrile-acidified with 0.1% (*v*/*v*) formic acid at a flow rate of 0.5 mL/min under gradient conditions: 0 min at 0% B, 2.5 min at 15% B, 10 min at 21% B, 12 min at 90% B, 13 min at 95% B, 15 min at 0% B, and 17 min at 0% B. Negative and positive ionization (ESI- and ESI+, respectively) was carried out with a capillary voltage of 2.0 kV and 3.5 kV, respectively. Mass spectra were acquired at a mass range of 50 to 1800 Da at low (6 eV) and high (15–45 eV) collision energy. Finally, a mass correction was performed with leucine-enkephalin (50 pg/mL), which was infused every 3 min. For polyphenol quantification, calibration curves were constructed with commercial standards. The linear regression models and validation parameters are included in the (Appendix A). The limits of detection and quantification (LOD and LOQ, respectively) were determined as 3 or 10 times the standard deviation of the intercept/slope.

### 4.4. Consumer Acceptability Evaluation

A consumer acceptability test was performed with an untrained panel of 50 participants (women and men from 21 to 55 years old who were consumers of fruit-based beverages) using an acceptance test. All berries-based beverages were served in clear glasses at 4 °C, and they were previously subjected to a microbiology analysis to assess their safety. Water was supplied as a palate cleanser between each sample. Panelists evaluated the color, aroma, flavor attributes, and overall acceptability according to a 9-point hedonic scale, where 9 = liked extremely and 1 = disliked extremely. Rejection was assumed if the acceptance index was <70% [20].

### 4.5. Stability Study

All berry-based beverages were stored in glass bottles in a constant climate chamber (HPP75 Memmert, Schwabach, Germany). At 4 °C, sampling was carried out at 0, 15, 30, 45, 60, 75, and 90 days, whereas sampling was carried out at 0, 3, 6, 9, and 12 days at 25 and 34 °C. The storage days for each temperature were established according to microbiology analyses carried out throughout the experiment, which indicated an increased count of mesophilic aerobic bacteria after 12 days of storage at 25 and 34 °C and after 90 days of storage at 4 °C. Physicochemical parameters and polyphenolic compound content were determined at each sampling time.

### 4.6. pH, Titratable Acidity, and Total Soluble Solids

The pH of berry-based beverages was determined using a pH meter (Horiba F-74BW). Titratable acidity was performed according to the 942.15 AOAC method, and the results were expressed as g eq. citric acid/100 mL. The total soluble solid content (°Brix) was determined using a digital refractometer (Hanna Instruments HI 96801) for AOAC (932.12) [21].

### 4.7. Color Measurement and Browning Index

Color parameters were measured at a wavelength range from 380 to 700 nm with a resolution of 5 nm. Data were processed in the ColorBySpectra software according to the CIE 1964 standard observer, the spectral distribution of the illuminant D65, and a viewing angle of 10°. The CIELAB parameters L*, a*, and b* were determined [22]. The color difference (ΔE) was calculated with the following equation:(1)$$\Delta E^{*}=\sqrt{{\left(L^{*}-L_{0}\right)}^{2}+{\left(a^{*}-a_{0}\right)}^{2}+{\left(b^{*}-b_{0}\right)}^{2}}$$
where L*, a*, and b* correspond to lightness, redness/greenness, and blueness/yellowness at the end of the storage time, while L_0_, a_0_, and b_0_ correspond to the beginning of the storage time, respectively. The browning index was determined as the ratio of the absorbances at 520 nm by the absorbances at 420 nm, as reported by Dorris et al. [13].

### 4.8. Quantification of Polyphenolic Compounds

#### 4.8.1. Total Monomeric Anthocyanins

The total monomeric anthocyanin content was determined by the pH differential method. Briefly, 50 µL of each sample was mixed with 175 µL of each buffer solution (0.025 M potassium chloride, pH 1, or 0.4 M sodium acetate, pH 4.5). Then, absorbances were measured at 510 and 700 nm [23]. Finally, results were expressed as the mg of cyanidin-3-*O*-glucoside/mL of the beverage using the following equations:(2)$$\mathrm{Total}\;\mathrm{absorbance}={\left(Abs_{510\;nm}-Abs_{700\;nm}\right)}_{pH\;1}-{\left(Abs_{510\;nm}-Abs_{700\;nm}\right)}_{pH\;4.5\;}$$
(3)$$\mathrm{Total}\;\mathrm{anthocyanins}=\frac{\mathrm{Total}\;\mathrm{absorbance}\times 1000\times MW\times DF}{\varepsilon \times d}$$
where Abs = absorbance, MW = molecular weight of cyanidin 3-*O*-glucoside (448.8 g/mol), DF = dilution factor, ε = molar extinction coefficient of cyanidin 3-*O*-glucoside (26,900 L mol/cm), and d = distance.

#### 4.8.2. Total Phenolic Compounds

The total polyphenol content was estimated according to Singleton et al. [24]. The samples had the following: 40 µL of distilled water, 25 µL of 1.0 N Folin Cicalteu reagent, and 125 µL of 20% Na_2_CO_3_. Then, the mixture was incubated for 30 min under darkness, and absorbances were measured at 765 nm. Total polyphenols were quantified with a calibration curve using gallic acid as the standard, and results were expressed as mg eq. gallic acid/mL.

#### 4.8.3. Total Proanthocyanidins

Total proanthocyanidins were determined with the vanillin-HCl methodology [25]. Samples (100 μL) were mixed with 1.2 mL of a 4% (*w*/*v*) vanillin solution in methanol. Then, 600 µL of HCl was added and stirred. Finally, the solution was incubated for 15 min under darkness, and absorbances were measured at 500 nm. Total proanthocyanidins were quantified using a calibration curve and expressed as mg eq. catechin/mL.

### 4.9. Model of Degradation Kinetics

First-order models were applied to fit the data of the total polyphenols, anthocyanins, and proanthocyanidins content of the beverages in relation to the effect of storage (degradation kinetics) [26]. The model parameters were identified using linear regressions on the logarithmic curves following the Arrhenius model from the following equation:(4)$$\ln\left(\frac{Y_{t}}{Y_{0}}\right)=-\mathrm{kt}$$
where Y_0_ and Y_t_ are the values of each response time during storage, t is the storage time (days), and k is the pre-exponential factor (s^−1^). Degradation rate constants were estimated, and half-life (t_1/2_) values were calculated with the following equation.
(5)$$t_{1/2}=\ln\left(2\right)/k$$

### 4.10. Statistical Analysis

The results were expressed as the mean ± standard deviation of three experimental replicates of the beverages and three technical replicates for each assay. Since all variables were parametric, as assessed by the Kolmogorov–Smirnov (data distribution) and Levene (homoscedasticity) tests, statistical analysis was determined by an analysis of variance (ANOVA) followed by Tukey’s test. Pearson’s correlation coefficient was used for the correlations between polyphenol degradation, color loss (CIELAB), browning index, and ΔE values. Statistical significance was considered at *p* ≤ 0.05. All statistical analyses were carried out with the JMP 11.0 statistical software package.

## 5. Conclusions

This study demonstrated that the elaboration of berry decoctions is a good alternative for the development of functional beverages due to their adequate sensorial acceptability and polyphenol content and profile. Interestingly, the strawberry–blueberry blend beverage showed increased sensory acceptability due to its color and flavor, as well as a high content and variety of polyphenols as compared to single-fruit-based beverages. Moreover, the blend berry-based beverage showed a longer polyphenol half-life time during storage, leading to a beverage with high color stability under commercial storage conditions. These results could be related to the integrative polyphenol profile of the blended berry-based beverage since some of these polyphenols could exert a copigmentation effect, decreasing their degradation rate and color loss. However, further studies must be conducted to understand the complex interactions between polyphenols and their impact on their stability and to evaluate their nutraceutical potential.

## Figures and Tables

**Figure 1 molecules-28-02496-f001:**
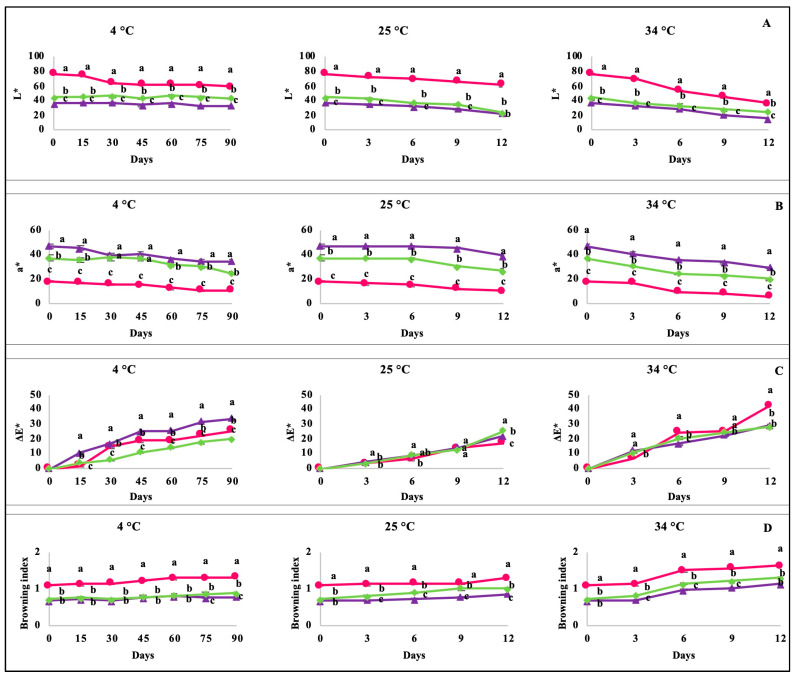
Changes in CIE parameters and the browning index of the berry-based beverages stored at 4 °C for 90 days and 25 °C and 34 °C for 12 days. (**A**) L*: Luminosity; (**B**) a*: redness/greenness; (**C**) ΔE: color difference; (**D**) browning index. Circles (●): Strawberry beverage; triangles (▲): blueberry beverages; diamond (♦): strawberry–blueberry beverage. Data are expressed as the mean ± standard deviation of three replicates. Different letters on the same day indicate significant (*p* < 0.05) differences between samples by Tukey’s test.

**Figure 2 molecules-28-02496-f002:**
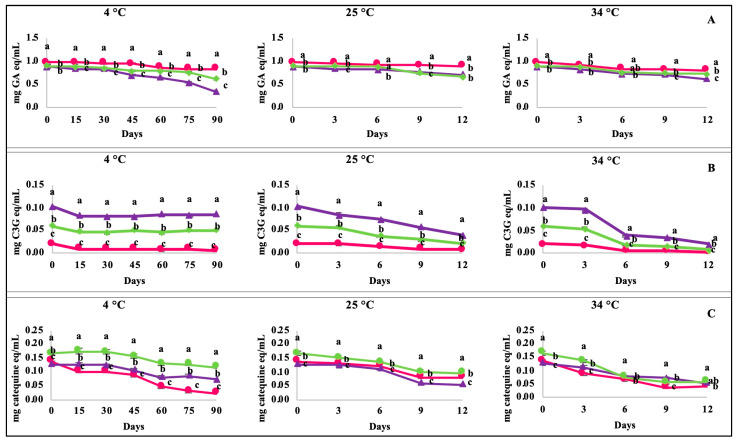
Degradation of polyphenolic compounds of the berry-based beverages at 4 °C, 25 °C, and 34 °C in storage. (**A**) Total phenolic compounds; (**B**) total monomeric anthocyanins; (**C**) total proanthocyanidins. Circles (●): strawberry beverage; triangles (▲): blueberry beverages; diamond (♦): strawberry–blueberry beverage. Data are expressed as the mean ± standard deviation of three replicates. Different letters on the same day indicate significant (*p* < 0.05) differences between samples by Tukey’s test.

**Figure 3 molecules-28-02496-f003:**
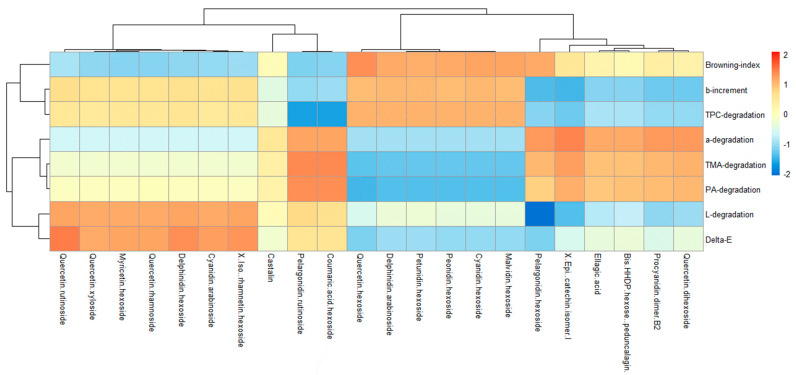
Correlation heatmap between the polyphenol profile of berry-based beverages and their color and polyphenol degradation during storage under commercial conditions. TPC: Total phenolic compounds; TMA: total monomeric anthocyanins; PA: proanthocyanidins.

**Table 1 molecules-28-02496-t001:** Identification of polyphenolic compounds of strawberry, blueberry, and strawberry–blueberry beverages by UPLC-ESI-Q-Tof-MS.

Compounds	Rt (min)	Molecular Formula	Expected Mass (Da)	Observed Mass (Da)	Mass Error (ppm)	Berry-Based Beverages		
						Strawberry	Blueberry	Strawberry–Blueberry
Anthocyanins								
Delphinidin hexoside	3.50	C_21_H_21_O_12_	465.1033	465.1017	−2.3598	ND	10.35 ± 0.15 ^a^	4.65 ± 0.84 ^b^
Delphinidin arabinoside	3.89	C_20_H_19_O_11_	435.0927	435.0900	−4.9877	0.01 ± 0.00 ^c^	4.93 ± 0.05 ^a^	2.27 ± 0.40 ^b^
Cyanidin hexoside	3.94	C_21_H_21_O_11_	449.1084	449.1089	2.4493	0.24 ± 0.06 ^c^	6.88 ± 0.09 ^a^	3.38 ± 0.51 ^b^
Petunidin hexoside	4.14	C_22_H_23_O_12_	479.1190	479.1221	7.7516	0.01 ± 0.00 ^c^	16.59 ± 0.16 ^a^	7.54 ± 1.53 ^b^
Cyanidin arabinoside	4.27	C_20_H_19_O_10_	419.0978	419.0969	−0.9809	ND	1.23 ± 0.17 ^a^	0.49 ± 0.15 ^b^
Pelargonidin hexoside	4.46	C_21_H_21_O_10_	433.1135	433.1134	1.0703	16.65 ± 3.72 ^a^	3.83 ± 4.38 ^a^	11.86 ± 0.44 ^a^
Malvidin hexoside	4.61	C_23_H_25_ClO_12_	493.1346	493.1347	1.3407	0.03 ± 0.00 ^c^	41.03 ± 0.10 ^a^	19.66 ± 3.48 ^b^
Peonidin hexoside	4.86	C_22_H_23_O_11_	463.1240	463.1235	0.0040	0.01 ± 0.00 ^c^	15.78 ± 0.88 ^a^	7.58 ± 2.02 ^b^
Pelargonidin rutinoside	5.07	C_27_H_31_O_14_	579.1714	579.1697	−1.9569	0.25 ± 0.00 ^a^	ND	0.14 ± 0.00 ^b^
Total						17.07 ± 3.61 ^c^	100.62 ± 3.45 ^a^	57.50 ± 9.26 ^b^
Flavanols								
Procyanidin dimer B2	3.45	C_30_H_26_O_12_	578.1424	577.1360	1.4800	0.48 ± 0.20 ^a^	0.12 ± 0.08 ^a^	0.24 ± 0.10 ^a^
(-)-Epicatechin	3.77	C_15_H_14_O_6_	290.0790	289.0709	−3.0812	0.56 ± 0.07 ^a^	0.02 ± 0.00 ^a^	0.22 ± 0.00 ^a^
Total						1.04 ± 0.27 ^a^	0.13 ± 0.06 ^b^	0.36 ± 0.05 ^b^
Flavonols								
(Iso)-rhamnetin hexoside	4.18	C_22_H_22_O_12_	478.1111	477.1052	2.8182	ND	0.33 ± 0.05 ^a^	0.11 ± 0.02 ^b^
Myricetin hexoside	5.19	C_21_H_20_O_13_	480.0904	479.0840	1.9089	ND	0.45 ± 0.03 ^a^	0.16 ± 0.09 ^b^
Quercetin rutinoside	5.69	C_27_H_30_O_16_	610.1534	609.1487	4.1921	ND	1.24 ± 0.05 ^a^	0.64 ± 0.03 ^b^
Quercetin hexoside	5.80	C_21_H_20_O_12_	464.0955	463.0902	4.3627	0.04 ± 0.00 ^b^	6.96 ± 0.18 ^a^	3.87 ± 0.81 ^ab^
Quercetin dihexoside	5.84	C_27_H_30_O_17_	478.0747	477.0678	0.7692	0.93 ± 0.46 ^a^	0.16 ± 0.00 ^a^	0.40 ± 0.17 ^a^
Quercetin xyloside	6.16	C_20_H_18_O_11_	434.0849	433.0788	2.6821	ND	0.41 ± 0.06 ^a^	0.13 ± 0.09 ^a^
Quercetin rhamnoside	6.40	C_21_H_20_O_11_	448.1006	447.0941	1.7316	ND	2.64 ± 0.01 ^a^	1.50 ± 0.39 ^a^
Total						0.95 ± 0.48 ^c^	12.12 ± 0.26 ^a^	6.81 ± 1.53 ^b^
Hydroxycinnamic acids								
Coumaric acid hexoside	4.02	C_15_H_18_O_8_	326.1002	325.0923	−1.9174	1.32 ± 0.03 ^a^	ND	0.47 ± 0.05 ^b^
Ellagic acid	5.60	C_14_H_6_O_8_	302.0063	300.9979	−3.7856	1.78 ± 1.30 ^a^	0.17 ± 0.21 ^a^	0.61 ± 0.61 ^a^
Total						3.10 ± 1.27 ^a^	0.18 ± 0.21 ^a^	1.08 ± 0.57 ^a^
Ellagitannins								
Bis-HHDP-hexose	1.65	C_34_H_24_O_22_	784.0759	783.0718	4.0607	1.72 ± 1.14 ^a^	0.32 ± 0.00 ^a^	0.62 ± 0.61 ^a^
Castalin	3.55	C_27_H_20_O_18_	632.0650	631.0602	3.9080	0.18 ± 0.11 ^a^	ND	0.11 ± 0.00 ^a^
Total						1.91 ± 1.25 ^a^	0.32 ± 0.00 ^a^	0.68 ± 0.69 ^a^

Data are shown as the mean ± standard deviation of three replicates. Results are expressed as μg/mL. Different letters within the same row differ significantly (*p* < 0.05) by Tukey’s test. Rt: Retention time. ND: Not detected.

**Table 2 molecules-28-02496-t002:** Mean value, skewness, and kurtosis of the consumer acceptability test carried out with a 9-point hedonic scale of berry-based beverages.

Sensory Characteristics	Beverages		
	Strawberry	Blueberry	Strawberry–Blueberry
Color			
Mean ± SD	7.24 ± 1.71 ^a^	7.82 ± 1.02 ^a^	7.80 ± 0.81 ^a^
Skewness	−0.98	−0.45	−0.58
Kurtosis	−0.03	−0.88	1.50
Aroma			
Mean ± SD	6.56 ± 1.75 ^a^	6.62 ± 1.41 ^a^	6.48 ± 1.54 ^a^
Skewness	−0.21	−0.55	−0.07
Kurtosis	−0.79	−0.52	−1.14
Flavor			
Mean ± SD	7.08 ± 1.87 ^a^	6.74 ± 1.54 ^a^	7.28 ± 1.28 ^a^
Skewness	−0.74	−0.63	−0.80
Kurtosis	−0.53	−0.03	0.66
Overall acceptability			
Mean ± SD	7.14 ± 1.59 ^a^	7.02 ± 1.22 ^a^	7.32 ± 1.15 ^a^
Skewness	−0.59	−0.60	−0.42
Kurtosis	−0.63	0.35	0.38

Data are shown as mean ± standard deviation (SD). Different letters within the same row differ significantly (*p* < 0.05).

**Table 3 molecules-28-02496-t003:** First-order reaction kinetics for total phenolic compounds, total monomeric anthocyanins, and total proanthocyanidins stability of the berry-based beverages stored at 4 °C for 90 days and 25 °C and 34 °C for 12 days.

Beverage	T	Total Phenols			Total Anthocyanins			Total Proanthocyanidins		
		*K* ^1^	t_1/2_ ^2^	*R*	*K* ^3^	t_1/2_ ^2^	*R*	*K* ^4^	t_1/2_ ^2^	*R*
**Strawberry**	4 °C	1.7 × 10^−3^ ± 0.0 ^f^	396.4 ± 16.0 ^a^	0.82	1.0 × 10^−2^ ± 0.0 ^f^	68.9 ± 0.4 ^c^	0.72	2.1 × 10^−2^ ± 0.0 ^e^	32.2 ± 0.4 ^c^	0.92
	25 °C	4.8 × 10^−3^ ± 0.0 ^e^	143.0 ± 6.2 ^c^	0.92	1.0 × 10^−1^ ± 0.0 ^c^	6.9 ± 0.1 ^de^	0.93	5.3 × 10^−2^ ± 0.0 ^d^	13.1 ± 0.7 ^d^	0.88
	34 °C	1.7 × 10^−2^ ± 0.0 ^c^	38.8 ± 2.7 ^e^	0.94	1.7 × 10^−1^ ± 0.0 ^a^	4.0 ± 0.0 ^e^	0.94	1.3 × 10^−1^ ± 0.0 ^a^	5.1 ± 0.7 ^d^	0.95
**Blueberry**	4 °C	9.3 × 10^−3^ ± 0.0 ^d^	74.2 ± 5.1 ^d^	0.84	3.3 × 10^−3^ ± 0.0 ^g^	210.0 ± 0.0 ^a^	0.80	7.0 × 10^−3^ ± 0.0 ^e^	98.9 ± 7.9 ^b^	0.91
	25 °C	1.7 × 10^−2^ ± 0.0 ^c^	39.5 ± 1.7 ^e^	0.97	7.6 × 10^−2^ ± 0.0 ^e^	9.0 ± 0.0 ^d^	0.96	7.6 × 10^−2^ ± 0.0 ^c^	9.0 ± 0.4 ^d^	0.84
	34 °C	2.6 × 10^−2^ ± 0.0 ^a^	26.5 ± 1.0 ^c^	0.91	1.3 × 10^−1^ ± 0.0 ^b^	5.1 ± 0.4 ^de^	0.96	6.4 × 10^−2^ ± 0.0 ^cd^	10.7 ± 0.6 ^d^	0.95
**Strawberry–Blueberry**	4 °C	3.6 × 10^−3^ ± 0.0 ^e^	190.2 ± 11.0 ^b^	0.82	5.4 × 10^−3^ ± 0.0 ^fg^	128.5 ± 6.7 ^b^	0.93	6.1 × 10^−3^ ± 0.0 ^e^	114.1 ± 10.5 ^a^	0.95
	25 °C	2.4 × 10^−2^ ± 0.0 ^a^	28.0 ± 0.6 ^e^	0.83	8.9 × 10^−2^ ± 0.0 ^d^	7.7 ± 0.1 ^de^	0.96	5.1 × 10^−2^ ± 0.0 ^d^	13.6 ± 0.3 ^d^	0.91
	34 °C	2.0 × 10^−2^ ± 0.0 ^b^	33.7 ± 1.2 ^e^	0.92	1.7 × 10^−1^ ± 0.0 ^a^	3.9 ± 0.1 ^e^	0.94	1.1 × 10^−1^ ± 0.0 ^b^	6.2 ± 0.4 ^d^	0.96

Data are shown as the mean ± standard deviation of three replicates. Different letters in the same column indicate significant (*p* < 0.05) differences between samples. T: temperature. ^1^ mg gallic acid eq/mL; ^2^ days; ^3^ mg cyanidin 3-*O*-glucoside eq/mL; ^4^ mg catechin eq/mL.

## Data Availability

The data presented in this research are available upon request from the corresponding authors.

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
