# Peer review of "Improved Phenolic Profile, Sensory Acceptability, and Storage Stability of Strawberry Decoction Beverages Added with Blueberry Decoctions"

_molecules, 2023, doi:10.3390/molecules28062496_

Round 1

Reviewer 1 Report

I have reviewed the manuscript entitled “Improved phenolic profile, sensory acceptability, and storage stability of strawberry decoction beverage added with blue-berry decoction”. In addition to the analytical methods, this article focuses on comparing the chemical and sensory characteristics of strawberry-blueberry blends in order to present data and argue that it is of academic and relevant industrial importance. From this point of view, this paper is good trial for analytical studies of fruit-based beverage.

However, I still have some suggestions for corrections, so please check it out.

1. In the introduction section, the research methods used in this paper need to be supplemented with appropriate literature.

2. Line 48-51: properly define the aim of work at the end of introduction. The aim of this study was to…

3. In the materials and methods, please provide source reference for each method used.

4. Line 283, please complete this sentence.

5. In the conclusions, in addition to summarizing the actions taken and results, please strengthen the explanation of their significance.

Author Response

Reviewer no.1

I have reviewed the manuscript entitled “Improved phenolic profile, sensory acceptability, and storage stability of strawberry decoction beverage added with blue-berry decoction”. In addition to the analytical methods, this article focuses on comparing the chemical and sensory characteristics of strawberry-blueberry blends in order to present data and argue that it is of academic and relevant industrial importance. From this point of view, this paper is good trial for analytical studies of fruit-based beverage. However, I still have some suggestions for corrections, so please check it out.

R. We appreciate the comment of the reviewer. We provide a point-by-point response to each comment made by the reviewer to improve the quality of our manuscript.

1. In the introduction section, the research methods used in this paper need to be supplemented with appropriate literature.

R. As suggested by the reviewer, we complemented the information included in the introduction section (Page 2, Lines 49-59). The following reference was included in the reference section (Page 13, Lines 490-491):

Kong, F., Singh, R.P. Chemical deterioration and physical instability of foods and beverages. In: Food and Beverage Stability and Shelf Life. Kilcast D., Subramaninam P., Eds., Woodhead Publishing Limited, UK, 2011; pp.29-56.

2. Line 48-51: properly define the aim of work at the end of introduction. The aim of this study was to…

R. As suggested by the reviewer, we modified the description of the aim of the study (Page 2, Lines 59-62).

3. In the materials and methods, please provide source reference for each method used.

R. As suggested by the reviewer, we revised the methodology section and noticed that two references were missing. Accordingly, the references [19] and [20] were added in the sections 4.4 Hedonic sensorial evaluation and 4.6 pH, titratable acidity, and total soluble solids, respectively (Page 11, Lines 362 and 377).

Durães, G.C.S., Pires, B.A.M., Lins, T.C.L. Kombucha based synbiotic beverage using Yacon (Smallanthus sonchifolius) as a fermentation substrate: development and sensorial analysis. Food Res. 2021, 5, 66-71.

AOAC International. Official Methods of Analysis of AOAC International; AOAC International: Gaithersburg, MD, USA, 2002.

These references were included in the reference section (Page 14, Lines 518-520.

4. Line 283, please complete this sentence.

R. We apologize for this mistake, we completed the sentence (Page 10, Lines 311-313).

5. In the conclusions, in addition to summarizing the actions taken and results, please strengthen the explanation of their significance.

R. We complemented the conclusion section with the following statement (Page 13, Lines 441-447).

Reviewer 2 Report

The authors mixed different berry materials to obtain a beverage with a mix of polyphenols. The authors should improve the manuscript prior to publication.

- Please include the objective of the study at the end of the introduction.

- In the results section, you should further describe how the compounds were identified, and the information shown in the table showing the identification.

- The hedonic test, should be better described as a consumer acceptability test.

- The way sensory evaluation data is shown is confusing, authors should add the mean value obtained in the hedonic scale with the different parameters evaluated.

- In the materials and method section, the chromatographic conditions (mobile phases, gradient, etc) should be detailed. 

- Similarly, the sensory evaluation test is barely described. Authors should include the number of consumers participating in the sensory evaluation test, the hedonic scale used, etc. 

Author Response

The authors mixed different berry materials to obtain a beverage with a mix of polyphenols. The authors should improve the manuscript prior to publication.

R. We appreciate the comment of the reviewer. We provide a point-by-point response to each comment made by the reviewer to improve the quality of our manuscript.

Please include the objective of the study at the end of the introduction.

R. As suggested by the reviewer, we modified the description of the aim of the study (Page 2, Lines 59-62).

In the results section, you should further describe how the compounds were identified, and the information shown in the table showing the identification.

R.  As suggested by the reviewer, we further described the results of the polyphenol profile of the berry-based beverages in the results section (Page 2, Lines 65-68 and 75-76; Page 3, Lines 82-85).

The hedonic test, should be better described as a consumer acceptability test.

R.  As suggested by the reviewer, this term was modified throughout the manuscript (Page 1, Lines 16-17; Page 2, Lines 57-58; Page 3, Line 93; Page 11, Lines 354 and 355) and in the title of Table 2 of the supplementary material.

The way sensory evaluation data is shown is confusing, authors should add the mean value obtained in the hedonic scale with the different parameters evaluated.

R.  The mean value and standard deviation of the 9-point score of each sensory parameter analyzed by the consumer acceptability test was originally included as supplementary material in Table S1, whereas their distribution (histograms) was included as Figure S1. To avoid further confusion as indicated by the reviewer, we exchanged these data, including the mean values, standard deviation, skewness, and kurtosis as Table 2 in the main document, whereas the histograms were included in the supplementary material as Figure S1.

In the materials and method section, the chromatographic conditions (mobile phases, gradient, etc) should be detailed.

R.  As suggested by the reviewer, we included the complete description of the LC-MS methodology (Page 10, Lines 335-353).

Similarly, the sensory evaluation test is barely described. Authors should include the number of consumers participating in the sensory evaluation test, the hedonic scale used, etc.

 R. As suggested by the reviewer, we included the complete description of the consumer acceptability test (Pages 10 and 11, Lines 355-362).

Reviewer 3 Report

The subject of the article was the analysis of drinks made of strawberries, blueberries and their mixtures. The polyphenol profile, a hedonic sensorial profile and stability analysis were carried out. The text is written in correct English, the structure is simple and clear. However, the article has many shortcomings and needs to be improved. Therefore, I recommend major revision. The comments on the article are given below:

1) Polyphenol profile by UPLC-ESI-Q-ToF MS of the berry-based beverage:

a) The experimental part does not contain any information regarding the quantitative analysis of the composition of beverages, e.g. calibration curves, method validation, etc.

b) Line 70 “On the other hand, the SB showed the highest concentration of hydroxycinnamic acids (3.1 μg/mL), such as coumaric acid hexoside and ellagic acid.” Blueberry is rich in phenolic acids, most prominently chlorogenic acid. https://pubs.acs.org/doi/10.1021/jf203812w)  Content of chlorogenic acids should be determined.

c) Accuracy and rounding of results – the errors are of the same order as the values. Maybe some calculations should be checked or measurements repeated?  Examples: SBB Peonidin hexoside 7.58 ± 2.02, BB  Pelargonidin hexoside  3.83 ± 4.38a

2) Hedonic sensorial evaluation - the number of participants in the study was not specified.

3) Table S3 / line 110. Nevertheless, the pH values significantly decreased after 12 days of storage at 25 and 34°C – Why? Difference is also visible in °Brix values.

4) Table S1. (Mean value, skewness, and kurtosis of the 9-point score hedonic sensory analysis of the berry-based beverage) - Taking into account the standard deviations, there were no differences in the sensory analysis of the drinks

5) Conclusion / Line 401 - Moreover, the blend berry-based beverage showed longer polyphenol half-life time during storage, leading to a beverage with high color stability under commercial storage conditions.  The trends of changes shown in Figure 3 do not support this conclusion. The mixture does not differ from the individual drinks in any of the measured parameters. Therefore, the conclusions should be corrected.

Author Response

The subject of the article was the analysis of drinks made of strawberries, blueberries and their mixtures. The polyphenol profile, a hedonic sensorial profile and stability analysis were carried out. The text is written in correct English, the structure is simple and clear. However, the article has many shortcomings and needs to be improved. Therefore, I recommend major revision. The comments on the article are given below:

R. We appreciate the comment of the reviewer. We provide a point-by-point response to each comment made by the reviewer to improve the quality of our manuscript.

1. Polyphenol profile by UPLC-ESI-Q-ToF MS of the berry-based beverage:

a) The experimental part does not contain any information regarding the quantitative analysis of the composition of beverages, e.g. calibration curves, method validation, etc.

R. As suggested by the reviewer, we included a detailed description of the LC-MS methodology used for the polyphenol profile analysis (Page 10, Lines 335-353). Additionally, we included the linear regression models and the validation parameters (LOD and LOQ) in Table S3 as supplementary material.

b) Line 70 “On the other hand, the SB showed the highest concentration of hydroxycinnamic acids (3.1 μg/mL), such as coumaric acid hexoside and ellagic acid.” Blueberry is rich in phenolic acids, most prominently chlorogenic acid. https://pubs.acs.org/doi/10.1021/jf203812w) Content of chlorogenic acids should be determined.

R.  In a previous study, we identified chlorogenic acid as the major phenolic acid of blueberry decoction (6.97 mg/mL; Reynoso-Camacho et al., 2021) using the same UPLC-ESI-QToF MSE method as used for this study. Therefore, we hypothesized that the absence of chlorogenic acid in the blueberry-based beverage could be due to its thermal degradation during the pasteurization process. This information was included in the discussion section (Page 8, Lines 238-242).

Reynoso-Camacho, R., Sotelo-González, A. M., Patiño-Ortiz, P., Rocha-Guzmán, N. E., & Pérez-Ramírez, I. F. Berry by-products obtained from a decoction process are a rich source of low-and high-molecular weight extractable and non-extractable polyphenols. Food Bioprod. Process, 2021,127, 371-387.

c) Accuracy and rounding of results – the errors are of the same order as the values. Maybe some calculations should be checked or measurements repeated? Examples: SBB Peonidin hexoside 7.58 ± 2.02, BB Pelargonidin hexoside 83 ± 4.38a

R.  We revised all raw data and the variation reported in Table 1 is correct. It is noteworthy that this analysis was carried out with true replicates, since beverages from three independent batches were analyzed. The high variability observed in unstable compounds like anthocyanins could be related to the decoction process, since dried fruits were extracted for 15 min in boiling water at 95 ºC.

2. Hedonic sensorial evaluation - the number of participants in the study was not specified.

R.  Fifty participants were included in the consumer acceptability assessment. This information was included in the methodology section as suggested by the reviewer (Page 11, Line 350-357).

3. Table S3 / line 110. Nevertheless, the pH values significantly decreased after 12 days of storage at 25 and 34°C – Why? Difference is also visible in °Brix values.

R.  We hypothesized that the citric acid included as an additive in the formulation for the beverages was not enough to prevent the growth of mesophilic aerobic bacteria after 15 days of storage at 25 and 34 ºC, which contribute to increasing acidity (lowering pH values) and decreasing sugar content. We included this statement in the discussion section (Page 9, Lines 266-271).

4. Table S1. (Mean value, skewness, and kurtosis of the 9-point score hedonic sensory analysis of the berry-based beverage) - Taking into account the standard deviations, there were no differences in the sensory analysis of the drinks

R.  Indeed, we clarify that no significant differences were observed in the description of these results (Page 3, Line 98; Page 4, Lines 105 and 115-116).

5. Conclusion / Line 401 - Moreover, the blend berry-based beverage showed longer polyphenol half-life time during storage, leading to a beverage with high color stability under commercial storage conditions. The trends of changes shown in Figure 3 do not support this conclusion. The mixture does not differ from the individual drinks in any of the measured parameters. Therefore, the conclusions should be corrected.

R.  Figure 3 shows the kinetic behaviors of the total polyphenol content of each berry-based beverage during storage, which does not show clear results in this regard as indicated by the reviewer. Nevertheless, when first-order reaction kinetics were obtained for each parameter, it was observed that the blend berry-based beverage showed higher polyphenol half-life values (Table 3), which was associated with a lower color loss (Figure 4).

Round 2

Reviewer 2 Report

The authors properly attended to my comments. I recommend acceptance of the manuscript in its present form. 

Reviewer 3 Report

The authors made the required changes. However, the argument about the cause of the thermal degradation of chlorogenic acid is questionable because anthocyanins have still been detected and they are less stable. Nevertheless, I recommend the article for publication.